Relationship of central incisor implant placement to the ridge configuration anterior to the nasopalatine canal in dentate and partially edentulous individuals: a comparative study

Jia Xueting
Hu Wenjie
Meng Huanxin kqhxmeng@126.com
Department of Periodontology, Peking University School and Hospital of Stomatology , Beijing , China
Levin Liran
Electronic publication date: 2015 Nov 3
Publication date: 2015
Volume: 3
Electronic Location ID: e1315
Received 2015 Jul 21; Accepted 2015 Sep 21
Copyright: © 2015 Jia et al.
Copyright year: 2015
Copyright holder: Jia et al.
License: This is an open access article distributed under the terms of the Creative Commons Attribution License, which permits unrestricted use, distribution, reproduction and adaptation in any medium and for any purpose provided that it is properly attributed. For attribution, the original author(s), title, publication source (PeerJ) and either DOI or URL of the article must be cited.
License URL: https://creativecommons.org/licenses/by/4.0/

Keywords: Alveolar bone, Anterior maxilla, Dental implants, Nasopalatine canal, Cone-beam computed tomography

Funding: National Ministry of Health 2010–2012 Capital Medical Development and Research Fund, PRC 2011-4025-04 This study was supported by key program of clinical specialty, National Ministry of Health (2010–2012), and Capital Medical Development and Research Fund, PRC (2011-4025-04). The funders had no role in study design, data collection and analysis, decision to publish, or preparation of the manuscript.

==============================
Background. The aims of this study were to investigate the ridge contour anterior to the nasopalatine canal, and the difference between the incidences of the nasopalatine canal perforation in dentate and partially edentulous patients by cone-beam computed tomography.

Methods. Cone-beam computed tomography scan images from 72 patients were selected from database and divided into dentate and partially edentulous groups. The configuration of the ridge anterior to the canal including palatal concavity depth, palatal concavity height, palatal concavity angle, bone height coronal to the incisive foramen, and bone width anterior to the canal was measured. A virtual implant placement procedure was used, and the incidences of perforation were evaluated after implant placement in the cingulum position with the long axis along with the designed crown.

Results. Comparing with variable values from dentate patients, the palatal concavity depth and angle were greater by 0.9 mm and 4°, and bone height was shorter by 1.1 mm in partially edentulous patients, respectively. Bone width in edentulous patients was narrower than in dentate patients by 1.2 mm at incisive foramen level and 0.9 mm at 8 mm subcrestal level, respectively. After 72 virtual cylindrical implants (4.1 × 12 mm) were placed, a total of 12 sites (16.7%) showed a perforation and three-fourths occurred in partially edentulous patients. After replacing with 72 tapered implants (4.3 × 13 mm), only 6 implants (8.3%) broke into the canal in the partially edentulous patient group.

Conclusions. The nasopalatine canal may get close to the implant site and the bone width anterior to the canal decreases after the central incisor extraction. The incidence of nasopalatine canal perforation may occur more commonly during delayed implant placement in central incisor missing patients.

Introduction

Dental implant restoration has become a very common treatment in dental practices (Chung et al., 2009; Fugazzotto, Vlassis & Butler, 2004; Scheller et al., 1998). In the esthetic zone, the primary goal of implant treatment is to re-establish both esthetics and function (Buser & Von Arx, 2000). As generally accepted, the implant placement is always based on a restorative-driven philosophy (Garber & Belser, 1995). According to this concept, the three-dimensional ideal implant position has been described (Buser, Martin & Belser, 2004; Funato et al., 2007; Grunder, Gracis & Capelli, 2005; Su et al., 2010; Tarnow, Cho & Wallace, 2000). Mesio-distally, a single implant should be at least 1.5 mm away from adjacent root surface (Buser, Martin & Belser, 2004; Grunder, Gracis & Capelli, 2005). Apico-coronally, the implant platform is supposed to be placed 2–4 mm apically to the designed mid-facial gingival margin (Buser, Martin & Belser, 2004; Funato et al., 2007). Bucco-lingually, the implant should be positioned slightly palatal to the incisal edge and 2 mm of buccal bone is recommended (Funato et al., 2007; Grunder, Gracis & Capelli, 2005). Regarding the optimal implant orientation, placement of an implant axis in alignment with the designed crown is recommended in order to fabricate a screw-retained implant crown and prevent from excessive off-axis loading (Chan et al., 2014).

However, in both immediate and delayed implant therapy, the nasopalatine canal (NPC) is often an anatomical limitation for a maxillary central incisor implant placement in an ideal position according to the restorative-driven philosophy. The NPC is a bony channel located posterior to the maxillary central incisors and connects the nasal floor with the oral cavity. The NPC contains the nasopalatine nerve, the terminal branch of descending nasopalatine artery, fibrous connective tissue, fat, and small salivary glands (Keith, 1979; Liang et al., 2009). The relative location of the NPC in the maxilla was previously described by assessing the dimension of the buccal bone plate anterior to this canal in some studies, and a proximity of the NPC to the implant surgical site after tooth extraction was reported (Bornstein et al., 2011; Mardinger et al., 2008; Tözüm et al., 2012). Moreover, placement of implants with invading into the NPC may lead to direct contact of the implant with connective tissue and cause a series of complications, including hemorrhage during operation, short term sensory disturbance postoperatively, non-osseointegration of implant and nasopalatine duct cyst formation (Casado et al., 2008; McCrea, 2014; Mraiwa et al., 2004; Peñarrocha et al., 2014; Takeshita et al., 2013). When the NPC interrupts the central incisor implant placement, more complicated surgical procedure is required and the risk of complications increases.

Through examination of computerized tomography images of 30 American patients, Kraut & Boyden (1998) studied the volumes of the NPC and bone anterior to the canal and reported that approximately 4% NPC will be detrimental to the implant placement. However, the incidence of perforation into the NPC is associated with not only the anatomic morphology, but also the feature of the implant and the three-dimensional implant position. The incidence of perforation into the NPC when a central incisor implant is placed in an ideal position following an optimal axis was not well known yet. In addition, the change of the ridge morphology caused by tooth loss may increase the incidence of perforation has not been assessed. Besides, the feature of the exposure and the risk factors of the perforation have never been analyzed. Whether a tapered implant or a minor adjustment of implant angulation could be beneficial for avoidance of perforation was also not well known.

Cone-beam computed tomography (CBCT) has been widely used in clinical evaluation before implant surgery because of the capability of accurate three-dimension imaging, relative low radiation dose and costs (White, 2008). Moreover, virtual implant placement in CBCT scan images could provide an overall evaluation of implant position and the proximity of the implant to the surrounding anatomic structure (Fortin et al., 2003; Katsoulis, Pazera & Mericske-Stern, 2009). The accuracy of computer-guided template-based implant dentistry was analyzed in previous literature (Schneider et al., 2009). The results showed that the mean horizontal deviation was approximately 1 mm at the entry point and approximately 1.6 mm at the apex. Although the reliability may be insufficient to perform a “blind” implant surgery considering the fine anatomic structure around the implant site, CBCT is still one of the most valuable imaging tools in dentistry. Reasonable accurate three-dimensional anatomic information and the relationship between implant and surrounding anatomical structures can be evaluated thoroughly by observing CBCT scans, and the likelihood of complications could reduce afterwards. The aims of this study were to investigate the ridge contour anterior to the NPC and the difference between the incidences of NPC perforation in dentate and partially edentulous patients by CBCT.

Materials and Methods

Patient selection

This study was approved by the Biomedical Ethics Committee of Peking University School of Stomatology (approval ID PKUSSIRB-201519006). The pre-existing CBCT (Vatech CT, Korea) data selected for this study were performed from January 2011 to July 2014 for treatment planning of implant procedures. Appropriate methodology and sample size were determined by a pilot study and power analysis. The sample size was calculated with α = 0.05 and power = 0.90. It was determined that a sample size of 36 specimens per group (for a total sample of 72) was needed to represent a clinically significant difference in bone width anterior to the NPC.

Patients and images selected for this study had to fulfill the following inclusion criteria: (1) Chinese adults with either full teeth of anterior maxillary sextant or missing one maxillary central incisor in this sextant; (2) the present maxillary anterior teeth without obvious crowding or spacing; (3) no deep (>3 mm) overbite or increased (>3 mm) overjet in the anterior teeth area; (4) at least two pairs of posterior teeth which could be retained with occlusal contact on each side; (5) complete CBCT scanning of premaxilla and NPC with clear images without scattering artifacts. Patients and images were excluded if: (1) both maxillary central incisor were present but the amount of alveolar bone loss exceeded one third of root length; (2) unhealed extraction sockets and the period after extraction was within 6 months; (3) any socket preservation procedures were performed in the missing maxillary central incisor sites; (4) both maxillary central incisors were missing; (5) alveolar ridge height of implant site was less than 14 mm or the ridge width was less than 3.5 mm at the level of 2 mm below the bone crest. Images were assigned into two groups: dentate and partially edentulous groups. When anterior maxillary sextant presented was classified as dentate group, while the edentulous ridge of missing one maxillary central incisor was classified as partially edentulous group. The distributions of age, gender, NPC shape on sagittal slice (Mardinger et al., 2008) and implant site were well matched between two groups, respectively.

Data reconstruction

All images were obtained using a CBCT machine (Vatech CT, Gyeonggi-do, Korea) with standardized routine procedures in the Peking University School of Stomatology by experienced radiologists. The imaging parameters were set at 90 kVp, 7.0 mAs, scan time 24 s, resolution 0.15 mm and a field of view that varied based on the region scanned. The scans included in this study were selected from the database and processed with a measurement software program (Ez3D2009 Premium Ver. 1.2.1.0) in a password-protected computer. The observer examined CBCT images using monitor at a 1,280 × 1,024 screen resolution under room lighting. The distance between display and the observer was approximately 30 cm. The scans were re-oriented so that the premaxilla was bilaterally symmetric and the long axis of the sagittal CBCT slice was determined following the long axis of the designed crown (connecting the bucco-lingual midpoint at the cemento-enamel junction and the point at the incisal edge) of the maxillary central incisor. The data were reconstructed with slices at an interval of 0.5 mm. The luminance and grayscale were adjusted to obtain clear CBCT views.

Configuration of ridge anterior to the NPC

The palatal concavity of the alveolar ridge anterior to the NPC was analyzed by examining the sagittal slices (Fig. 1A) and measuring:

(1) The palatal concavity depth (PCD), the distance between the deepest point of the buccal plate on the palatal side and a reference line parallel to the sagittal long axis of the central incisor crown and passing through the labial opening of incisive foramen.

(2) The palatal concavity height (PCH), the distance between the deepest point of the buccal plate on the lingual side and a reference line perpendicular to the sagittal long axis of the central incisor crown and passing through the alveolar bone crest.

(3) The palatal concavity angulation (PCA), the angulation between the line connecting the deepest point of the palatal concavity and the labial opening of incisive foramen and the line parallel to the long axis of the central incisor crown and passing through the deepest point of the palatal concavity.

Figure 1 Configuration of ridge anterior to the NPC.

(A) a, the palatal concavity depth (PCD); b, the palatal concavity height (PCH); c, the palatal concavity angulation (PCA); d, the height of the alveolar bone coronal to the NPC (BH). (B) The arrow stands for minimum bone width anterior to the NPC (BW), measured at incisive foramen level, 8 mm and 14 mm below bone crest level, respectively.

In addition, the height of the alveolar bone coronal to the NPC (BH) was also recorded by measuring the vertical distance between the alveolar bone crest and the line perpendicular to the sagittal long axis of the central incisor crown and passing through the labial opening of incisive foramen in the midsagittal plane of the NPC (Fig. 1A).

The minimum width of buccal bone plate anterior to the NPC (BW) was measured in the axial view images at three levels: incisive foramen level, 8 mm subcrestal level and 14 mm subcrestal level (Fig. 1B).

Relative location of the NPC and the virtual implant

Seventy-two cylindrical implants (Straumann Bone-Level Implant, 4.1 × 12 mm, Fig. 2) and 72 tapered implants (Nobel Replace Tapered Implant 4.3 × 13 mm, Fig. 2) were placed virtually in the selected maxillary central incisor sites sequentially.

Figure 2 Features of the selected implants.

(A) Straumann Bone-Level Implant 4.1 × 12 mm; (B) Nobel Replace Tapered Implant 4.3 × 13 mm.

In the dentate group, each implant was placed in the midsagittal plane of selected maxillary central incisor mesio-distally. Bucco-lingually, the most lingual point of the implant platform was located at the cingulum of the central incisor. Apico-coronally, the implant platform was placed 2 mm below the crestal level. The sagittal long axis of the implant was parallel to the central incisor crown (Figs. 3A–3C).

Figure 3 Three-dimensional location of virtual implant in dentate patients (A–C) and partially edentulous patients (D–F).

(A) and (D) mesio-distal location in dentate and partially edentulous group; (B) and (E) bucco-lingually, the implant platform was placed at the cingulum of the future restoration in both dentate and partially edentulous group; (C) and (F) apico-coronally, the implant platform was located 2 mm below the alveolar bone crest in both groups.

In the partially edentulous group, each implant was placed in the center of the edentulous site mesio-distally. Bucco-lingually, the implant platform was also placed at the cingulum region (Chan et al., 2014). The details were present as follows: connecting the most prominent points of the two lateral incisors or the two canines on their palatal side to draw a reference line and measuring the distance between the cingulum of the natural contralateral central incisor at its most palatal point and the reference line, and then the most palatal point of the implant platform was placed labial to the reference line by the same distance. Apico-coronally, the location of the implant platform was the same as that of the dentate group. The sagittal long axis of the implant was parallel to the contralateral central incisor crown (Figs. 3D–3F).

After each virtual implant was placed, whether the implant penetrating through the interior wall of the NPC was assessed in the sagittal and axial views slice by slice. For the NPC perforation cases caused by cylindrical implants, the position of the implant platform was kept unchanged and the embedded direction was rotated distally and labially by a minor angulation (5°and 10°), respectively. The size and the location of each perforation were measured in the sagittal and axial view images, which included its length, depth, area and the distance between the most coronal point of the perforation and the alveolar bone crest (Fig. 4).

Figure 4 Description of nasopalatine canal (NPC) perforation in both sagittal slice and axial slice.

a, length of exposure; b, distance between the alveolar crest and the perforation (location of perforation); c, the depth of the exposure; d, the area of the exposure.

All measurements were conducted by two examiners (XJ and WH). The inter- and intraexaminer agreement was determined by comparing two repeated measurements at 20 randomly chosen sites taken 1-week apart.

Statistical analysis

All statistical analysis was performed using a statistical package (IBM, SPSS Statistics 19.0). The inter- and intraexaminer agreement was determined using a t test. All measurements were presented as means ± standard deviations (SDs). The occurrence of the NPC perforation was expressed as the number of sites and the percentage of the number of sites divided by the total number of sites. The PCD, PCH, PCA, BH and BW were compared between dentate group and partially edentulous group by Mann–Whitney U test. The chi-square test was used to compare the incidences of perforation between groups, genders and sides. Univariate and multivariate logistic regression analyses were performed to identify risk factors associated with the NPC perforation with the significance level at α = 0.05.

Results

A total of 703 subjects were screened and 72 subjects (54 males and 18 females) were selected for this study. The mean age was 45.6 years, with a range of 28–64 years of age. Each group consisted of 36 subjects. The age, gender and implant site were well matched between the dentate group and the partially edentulous group, respectively. The distribution of NPC shape recorded on sagittal plane did not show statistically significant differences between groups (Table 1). The intra-examiner and inter-examiner agreements were 0.94 and 0.87, respectively (p > 0.05).

Table 1 General characteristics of tested subjects.

	Total (n = 72)	Dentate group (n = 36)	Partially edentulous group (n = 36)	p value	
Age (mean ± SD)	45.6 ± 8.8	45.5 ± 9.0	45.6 ± 8.8	0.906	
Gender (n)					
Male	54	27	27		
Female	18	9	9	1.000	
Implant site (n)					
Right central incisor	30	15	15		
Left central incisor	42	21	21	1.000	
Canal shapes (n)					
Cylindrical	29	17	12		
Funnel-like	17	6	11		
Hourglass-like	15	9	6		
Banana-like	11	4	7	0.290	

The measuring results of configuration of ridge anterior to the NPC are shown in Table 2. A total of 54 ridges (75.0%) showed a palatal concavity in sagittal views. The mean and SD of PCD, PCH and PCA values were 1.8 ± 1.7 mm, 14.3 ± 7.3 mm and 8.6 ± 6.5°, respectively. The incisive foramen was located at 5.9 ± 2.3 mm below the alveolar bone crest. The mean and SD values of BWs at the incisive foramen level, 8 mm and 14 mm related to the subcrestal level were 6.0 ± 1.7 mm, 6.3 ± 1.5 mm and 6.9 ± 1.9 mm, respectively. Results of comparisons of ridge configuration between the dentate group and the partially edentulous group were also listed in Table 2 and shown in Fig. 5. There was statistically significant difference between the mean PCD values of the dentate group and the partially edentulous group (1.4 ± 1.4 mm vs. 2.3 ± 1.9 mm, p = 0.036). In addition, the mean PCA values of the dentate group and the partially edentulous group were 6.6° and 10.6°, respectively (p = 0.022). The distance between the incisive foramen and the bone crest was significant closer in the partially edentulous group than in the dentate group by 1.1 mm approximately (p = 0.022). At the incisive foramen level, the mean BW was statistically significantly thinner in the partially edentulous group than the dentate group, 5.4 ± 2.5 mm and 6.6 ± 1.1 mm, respectively (p = 0.013). Furthermore, the mean BW values measured at 8 mm subcrestal level of the dentate group and the partially edentulous group were 6.7 mm and 5.8 mm, respectively (p = 0.028). There was no statistically significant difference between groups of the PCH and BW values at 14 mm subcrestal level (p > 0.05).

Figure 5 Comparison of ridge configuration anterior to the nasopalatine canal (NPC) between dentate and partially edentulous patients.

The red line stands for the ridge contour of partially edentulous patients.

Table 2 Ridge configuration and comparison results (mean ± SD).

Variable		Dental status	Gender	
	Total (n = 72)	Dentate (n = 36)	Partially edentulous (n = 36)	p	Male (n = 54)	Female (n = 18)	p	
Palatal concavity								
PCD (mm)	1.8 ± 1.7	1.4 ± 1.4	2.3 ± 1.9	0.036	1.9 ± 1.7	1.6 ± 1.8	0.349	
PCH (mm)	14.3 ± 7.3	13.9 ± 7.8	14.8 ± 6.8	0.539	14.9 ± 7.7	12.6 ± 5.8	0.275	
PCA (°)	8.6 ± 6.5	6.6 ± 5.9	10.6 ± 6.5	0.022	8.4 ± 6.3	9.1 ± 7.2	0.758	
BH (mm)	5.9 ± 2.3	6.5 ± 1.9	5.4 ± 2.5	0.022	5.9 ± 2.4	5.8 ± 1.7	0.958	
BW (mm)								
Incisive foramen level	6.0 ± 1.7	6.6 ± 1.1	5.4 ± 2.0	0.013	6.2 ± 1.7	5.4 ± 1.6	0.040	
8 mm subcrestal level	6.3 ± 1.5	6.7 ± 1.2	5.8 ± 1.6	0.028	6.5 ± 1.4	5.5 ± 1.4	0.004	
14 mm subcrestal level	6.9 ± 1.9	7.0 ± 1.6	6.7 ± 2.1	0.401	7.1 ± 2.0	6.4 ± 1.3	0.141	
Notes.

PCD palatal concavity depth

PCH palatal concavity height

PCA palatal concavity angulation

BH bone height coronal to the canal

BW bone width anterior to the canal

Table 2 also illustrated the comparison results between genders. At the incisive foramen level, the mean BW was statistically significantly greater in male subjects compared with female subjects by 0.8 mm (p = 0.040). In addition, the mean BWs measured at 8 mm subcrestal level of the male and the female subjects were 6.5 mm and 5.5 mm, respectively (p = 0.004). The PCD, PCH, BH values were greater in male subjects, although the statistically significant difference did not exist (p > 0.05). When virtual cylindrical implants (4.1 × 12 mm) were placed, a total of 12 sites (16.7%) showed invading and perforation (Table 3). Three cases of them occurred in the dentate group (8.3%) while other nine cases occurred in the partially edentulous group (25.0%). The incidence of perforation was much higher in the partially edentulous group, although the statistically significant difference did not exist (p = 0.058). With respect to the implant site, the incidence of perforation was statistically significantly higher in the right central incisor site than in the left central incisor site, 33.3% and 4.8%, respectively (p = 0.001). The occurrence of perforation did not show statistically significant differences between genders (p > 0.05). In the axial view images, all the perforations were located at the mesio-palatal site of the virtual implant. Furthermore, the depth and the area of exposure were 0.7 ± 0.6 mm (range = 0.2–2.1 mm) and 1.0 ± 1.3 mm2 (range = 0.2–4.7 mm2), respectively. In the sagittal view images, the exposure located at 8.5 ± 3.5 mm below the alveolar bone crest, with a range of 2.3 mm–12.1 mm, and the length of the exposure was 5.1 ± 3.4 mm, with a range of 1.6 mm–12.0 mm.

Table 3 Frequency distribution of perforation with different implant type.

Group	Implants (number)	Number of perforations (number and percent)	
		Cylindrical 4.1 × 12 mm	Tapered 4.3 × 13 mm	
Dental status				
Dentate group	36	3 (8.3%)	0 (0.0%)**	
Partially edentulous group	36	9 (25.0%)	6 (16.7%)**	
Implant site				
Right central incisor site	30	10 (33.3%)*	5 (16.7%)***	
Left central incisor site	42	2 (4.8%)*	1 (2.4%)***	
Gender				
Male	54	7 (13.0%)	3 (5.6%)	
Female	18	5 (27.8%)	3 (16.7%)	
Total	72	12 (16.7%)	6 (8.3%)	
Notes.

* Statistically significant difference exists between implant sites with cylindrical implant (p = 0.001).

** Statistically significant difference exists between dentate and partially edentulous groups with tapered implant (p = 0.011).

*** Statistically significant difference exists between implant sites with tapered implant (p = 0.031).

After replacing the cylindrical implants with the tapered implants (4.3 × 13 mm), a total of 6 implants (8.3%) entered into the NPC, which all belonged to the partially edentulous group (Table 3). The incidence of perforation with the selected tapered implant was statistically significantly different between the dentate and partially edentulous groups (p = 0.011). Besides, five out of six perforations occurred in the right central incisor sites, and the statistically significant difference existed between different sides (p = 0.031). The location of the exposure was at the 6.2 ± 3.2 mm below the alveolar bone crest. The length, depth and area of the exposure were 5.4 ± 3.1 mm, 0.8 ± 0.6 mm and 1.3 ± 1.7 mm2, respectively.

The numbers of perforation sites was reduced to 4 (5.6%) and 2 (2.8%) by tilting the embedded direction of the cylindrical implant distal-apically by 5° and 10°, respectively. After the embedded direction was rotated labial-apically by the same degrees (5° and 10°), the incidence of perforation decreased to 8.3% and 4.2%, respectively. The changes of incidences of perforation, as well as the features of exposure after a minor adjustment of cylindrical implant angulation, were presented in Table 4.

Table 4 Incidence of perforation after implant angulation adjustment.

Embedded direction	Number (%)	Location (mm)	Length (mm)	Depth (mm)	Area (mm2)	
Axis of restoration	12 (16.7%)	8.5 ± 3.5	5.1 ± 3.4	0.7 ± 0.6	1.0 ± 1.3	
Distal by 5°	4 (5.6%)	5.4 ± 3.7	5.5 ± 2.0	0.8 ± 0.4	3.9 ± 4.9	
Distal by 10°	2 (2.8%)	2.4 ± 0.1	6.6 ± 2.1	1.1 ± 0.2	2.3 ± 0.7	
Labial by 5°	6 (8.3%)	6.8 ± 3.7	5.9 ± 3.9	0.9 ± 0.7	1.5 ± 1.9	
Labial by 10°	3 (4.2%)	5.3 ± 4.9	8.2 ± 5.1	1.2 ± 0.9	2.4 ± 2.4	

The multivariate logistic regression analysis revealed that the PCD was a statistically significant risk factor of perforation (OR 4.332; 95% CI [1.596–11.760]; p = 0.004). Implant placement in the left central incisor site (OR 0.087; 95% CI [0.010–0.783]; p = 0.029) and BW measured at 8 mm below the alveolar bone crest (OR 0.273; 95% CI [0.111–0.671]; p = 0.005) were two protective factors appeared in the last model (Table 5).

Table 5 Multivariate logistic regression of factors affecting the NPC perforation.

Variables	β	OR	95% CI	p value	
Left central incisor implant site	−2.446	0.087	0.010–0.783	0.029	
BW measured at 8 mm below crest	−1.299	0.273	0.111–0.671	0.005	
PCD	1.466	4.332	1.596–11.760	0.004	
Constant	1.974	7.200		0.329	
Notes.

NPC nasopalatine canal

BW bone weight anterior to the canal

PCD palatal concavity depth

Discussion

Chan and colleagues (2014) used CBCT imaging technique and found that a buccal concavity of ridge always existed anterior to the maxillary central incisor. The mean value of buccal concavity depth was reported to have mean of 3.42 mm, and it was associated with the occurrence of buccal plate fenestration. However, few studies have provided information regarding the palatal concavity and its relationship with the NPC during implant placement procedure. In the present study, the perforations of the NPC by the virtual implants were located at the labia-distal side only. Therefore, with an obvious palatal concavity, that means the NPC is located relatively at the labial side, may increase the risk of implant entering and damaging the neurovascular bundles within the NPC. In this study, 75% of ridges were present with a palatal concavity. More importantly, the palatal concavity depth was a statistically significant risk factor of NPC perforation. Therefore, not only the location of the incisive foramen, but also the trend of the NPC configuration should be carefully evaluated by CBCT during diagnostic procedure and treatment plan for implant therapy.

Bone dimensions anterior to the NPC are important factors for successful implant placement. In previous studies, bone dimensions were measured at crestal, middle, and (or) the most apical point of the canal in the midsagittal plane of the NPC with the reference line perpendicular to the maxillary plane or the sagittal long axis of the canal (Bornstein et al., 2011; Tözüm et al., 2012). A mean bone width of 7.17 ± 1.49 mm has been reported in a multicenter study (Tözüm et al., 2012). However, the implant is rarely placed in the midsagittal plane of the NPC, and also not involving the nasal part of the canal. In addition, the embedded direction may be different from the direction of measurement mentioned above. As a result, the data obtained by previous measuring methods might not reflect the implant condition accurately. In this study, the bone width anterior to the NPC was first measured in the axial view images at three levels: the incisive foramen level, 8 mm below the alveolar bone crestal level, and 14 mm below the crestal level. The incisive foramen level is where the NPC prevents the implant from placement procedure at the early beginning. The 8 mm and 14 mm below the bone crestal levels may represent the middle level and the apex level of the virtual implant selected in this study, respectively. In addition, the measuring direction was perpendicular to embedded direction of implant, that is, the sagittal long axis of the restoration. As a result, the measuring results of the present study would reflect the real implant condition with better accuracy. In the present study, the mean bone width anterior to the NPC was 6.0 and 6.3 mm at the incisive foramen level and 8 mm below the alveolar crest level, respectively. The results were slightly narrower than the bone width (7.17 mm) reported by Tözüm et al. (2012).

The incidence of NPC perforation during the maxillary central incisor implant procedure was evaluated using virtual implant placements in CBCT images. Using a cylindrical central incisor implant (4.1 × 12 mm) placed in the cingulum position with the long axis following that of its restoration, the incidence of NPC perforation was revealed to be 16.7% and significant higher than a previous study reported in using computerized tomography scanning images of American patients (Kraut & Boyden, 1998). The increased versatility of both latest CBCT and software could provide a more accurate and precise measurement of the relationship between implant and anatomical structures. Considering that the incidence of NPC perforation is associated with not only the anatomic morphology, but also the feature of implant and the three-dimensional implant position, the results present in this study may reflect the clinical implant condition more accurately. In addition, different racial sampling may have some effects on the skeletal development. Another study of our research team showed that the mean closest distance between the NPC and the apex of the central incisor root were 3.88 mm in axial CBCT images (X Jia et al., 2015, unpublished data), much closer than the mean distance of 5.22 mm reported by Chatriyanuyoke et al. (2012). The relatively lower values of the closest distances implied that insertion of implants into the NPC might be more likely to occur in Chinese patients.

The absence of maxillary central incisors affected some dimension changes of bony structure and incidences of NPC perforation. The results of comparison between the dentate and partially edentulous group revealed that, the PCD and PCA were statistically significantly greater in the partially edentulous group by 0.9 mm and 4°, respectively, although the distribution of NPC shape recorded on sagittal plane did not show statistically significant differences between groups. Mardinger et al. (2008) also found that the bucco-lingual NPC diameter was wider along the degree of ridge resorption. In the present study, it is implied that a closer proximity of NPC to implant site might be presented after tooth loss for a while with wound healed. In addition, bone width anterior to the canal and the bone height coronal to the canal were greater in dentate subjects in the present study by 1.2 mm and 1.1 mm respectively, mainly due to the alveolar bone remodeling after tooth loss (Araújo & Lindhe, 2005; Schropp et al., 2003). Other studies reported similar results regarding the change of bone width after tooth loss as the present study (Mardinger et al., 2008; Tözüm et al., 2012). Considering the ridge modeling after tooth loss, including the change of PCD, PCA, BH and BW as mentioned before, it would be no surprise that the incidence of NPC perforation was significantly higher in the partially edentulous group than the dentate group (25.0% and 8.3% after cylindrical implant placement; 16.7% and 0.0% after tapered implant placement). It is indicated that delayed implant placement in the maxillary central incisor site may require more care to avoid NPC perforation (Table 3). Another thing to keep in mind is that, in this study, one maxillary central incisor was present in the partially edentulous group for reference. Considering that loss of both central incisors may affect the ridge resorption, the potential of NPC perforation during delayed implant placement might increase in patients without both maxillary central incisors.

The results showed that the gender influenced the alveolar bone dimensions anterior to the NPC (Table 2). The mean BW values measured at the incisive foramen level and at 8 mm subcrestal level were significantly greater in men compared with women by 0.8 mm and 1.0 mm, respectively (p < 0.05). Our results correlates well with some other studies. Güncü et al. (2013) reported that buccal bone thickness were greater in male subjects. Chatriyanuyoke et al. (2012) also found that men had larger amounts of bone between the NPC and the MCIR sockets than women. Considering the results of comparisons of anatomic structures between genders, it would not be a surprise that insertion of implants into the NPC was more likely to occur in women, although the difference of incidence was not statistically significant, maybe due to insufficient sample size.

Another interesting finding was that the perforation usually occurred in the right central incisor site. The multivariate logistic regression analysis also revealed that the implant site was associated with the occurrence of NPC perforation (Table 5). This corresponds with the results of our preliminary study about the location of the NPC leaning on the right side at both the incisive foramen level and the apical level (X Jia et al., 2015, unpublished data).

The location, length, depth, and area of perforation are important information for implant placement. In the axial view images, all the perforations were located at the mesio-palatal side of the implant. However, in the sagittal view images, the perforation could occur at any part of implant (2.3–12.1 mm below the bone crest). The mean distance between the exposure and the crest was 8.5 mm in the present study, which meant that the perforation usually occurred at the mid-root level of the implant. A mean length of exposure of 5.1 mm indicated that the NPC perforation could not be ignored. However, on the other hand, the depth of exposure was only 0.7 mm on average, which meant that a tapered implant or a minor adjustment of implant angulation might be beneficial to prevent NPC from being invaded. In this study, tapered implant platform (4.3 × 13 mm) was selected and larger than the cylindrical implant in diameter by 0.2 mm. However, the diameter of the tapered implant will be narrowed to 4.1 mm at about 3.4 mm below the implant platform level, and only proximately 2.56 mm in diameter at the implant apical level, narrower than the cylindrical implant by approximately 1.5 mm (Fig. 2). Considering the relative shallow depth of exposure after cylindrical implant placement, a significant decrease of the incidence of perforation from 16.7% to 8.3% will be achieved if a cylindrical implant is replaced with a tapered implant (4.3 × 13 mm). A minor change of embedded direction was also beneficial for decreasing NPC perforation percentage from 16.7% down to 8.3% to 2.8% (Table 4). However, before the adjustment of implant angulation, practitioners should keep the proximity of adjacent lateral incisor to the implant site and the existing buccal concavity in mind. The distance between the implant apex and the adjacent root surface should not be too close and should not be less than 1.5 mm separation after rotating the implant to the distal, while the buccal plate fenestration is always the major factor to be considered during rotating the implant to the labial.

However, neither selected a tapered implant nor a minor adjustment (less than 10°) of implant angulation can avoid NPC perforation successfully in some cases. Therefore, it is recommended to take full analysis of the NPC using CBCT at the time of implant treatment planning with consideration of individual differences. The results in this study suggested that other appropriate features of implant, for example a shorter implant or a narrower implant, or a greater embedded angle that departed from the axis of the restoration might be selected to avoid perforation in some cases. For the cases that the implant may invade into the NPC inevitably, the debridement or the displacement of the neurovascular bundle in conjunction with the guided bone regeneration were proposed to prevent direct contact of implant surface with the neurovascular bundle and to provide adequate bone (Artzi et al., 2000; Peñarrocha et al., 2014; Rosenquist & Nyström, 1992; Verardi & Pastagia, 2012). These methods could significantly improve the condition of implant placement, but more long-term studies with large samples are necessary.

The limitation of this study is that the results are based on virtual analysis. When taking the findings of this study for reference, the practitioners should keep some consideration in mind, including the accuracy of CBCT scans, the features of the implant, and the surgical experience. Further clinical trial is required to clarify this issue. In addition, unassisted socket healing cases were adopted in the present study, although alveolar ridge preservation has become a key component of contemporary clinical dentistry (Avila-Ortiz et al., 2014). However, the effects of socket grafting and preservation procedures on the contour changes of the NPC are almost unknown. There is a need to evaluate the impact of specific alveolar ridge preservation procedures on the contour changes of the NPC and perforation incidences during implant placement in future.

Conclusions

Within the limits of this study, it can be concluded that the NPC may get close to the implant site after the central incisor extraction, and the bone width anterior to the canal may also reduce. The NPC perforations may occur more commonly in partially edentulous patients for delayed implant placement and in the right central incisor site. The right central incisor site with narrower bone width measured at 8 mm below the crest and a deep palatal concavity are associated with the common occurrence of NPC perforation during implant placement. A minor adjustment of implant angulation or using a tapered implant may be beneficial for preventing from NPC perforation.

Supplemental Information

Supplemental Information 1 Raw data of the present study

Click here for additional data file.

Additional Information and Declarations

Competing Interests

Author Contributions

Human Ethics

The authors declare there are no competing interests.

Xueting Jia, Wenjie Hu and Huanxin Meng conceived and designed the experiments, performed the experiments, analyzed the data, contributed reagents/materials/analysis tools, wrote the paper, prepared figures and/or tables, reviewed drafts of the paper.

The following information was supplied relating to ethical approvals (i.e., approving body and any reference numbers):

Biomedical Ethics Committee of Peking University School of Stomatology (approval ID PKUSSIRB-201519006).

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
