# Peer review of "Relationship of central incisor implant placement to the ridge configuration anterior to the nasopalatine canal in dentate and partially edentulous individuals: a comparative study"

_PeerJ, doi:10.7717/peerj.1315_

## Round 0.1 · original submission · Major Revisions

Please revise carefully according to the reviewers' comments and provide a detailed rebuttal letter describing the revisions that were performed.

Reviewer 1 ·

Basic reporting

see the attached letter

Experimental design

see the attached letter

Validity of the findings

see the attached letter

Annotated reviews are not available for download in order to protect the identity of reviewers who chose to remain anonymous.

Reviewer 2 ·

Basic reporting

1. The aims of the study are not compatible with the findings. Lines 79-82: "The aims of the this study are to investigate the contour alteration of the ridge..."- The study did not evaluate changes in the contour due to its retrospective nature. In addition it did not assess the dimensions prior to and following tooth extraction. Therefore, the aims should be re-defined.

2. The term "dentulous" should be changed. Patients may alternatively be classified as either dentate or partially/fully edentulous.

3. In the abstract the terms "lingual" should be changesd to "palatal" since it is the upper jaw.
The word "were" should be changed to "was": "The configuration of the ridge anterior to the canal were..."

4. Line 63: Additional complications that may result from NPC perforations should be discussed, such as nerve injury and bleeding.

5. Line 54: "in both implants and delayed implant therapy..."- Unclear. Did you mean immediate and delayed implants?

6. Lines 48-51: You mentioned guidelines for implant placement. A reference should be noted.

Experimental design

1. You mentioned in the discussion that the placement of an implant should be determined based on the prosthetic needs. Therefore, the placement of the implant on the CBCT slides could have been performed according to an artificial tooth that can be placed in implants planning software instead of according to artificial lines between the adjacent teeth.

2. The time following extraction is not mentioned. It may be detrimental for alveolar ridge remodeling and should be mentioned.

3. Alveolar bone dimensions anterior to the NPC should be compared between males and females.

Validity of the findings

1. The study compared the incidence of perforations when placing cylindrical versus tapered implant. A figure showing the difference between both implants should be added.

2. The article should discuss more thoroughly the importance of avoiding such perforations to the NPC (bleeding etc...)- A paper that can be used and cited was written by Guliz N. Guncu et al. COIR September 2103 p. 1023 " Is there a gender difference in anatomic features od incisive canal and maxillary environmental bone?"

3. The correlation between examiners is written in the results, however, it is not mentioned in the statistical analysis in the "Materials and Methods"- Which statistical test did you use in order to determine inter and inter- examiner agreement?

Additional comments

The language should be improved

Reviewer 3 ·

Basic reporting

the aims of this study are to investigate the contour alteration of the ridge anterior to the nasopalatine canal after maxillary tooth loss in comparison with dentulous patient and the difference between the incidences of the nasopalatine canal perforation in dentulous and edentulous patients by cone-beam computed tomography.
I find this topic new but does not have very important clinical relevance.
the introduction is too short, I expect them to explain more about the disadvantages and the limitation of CBCT

Experimental design

please explain why the measurement is significant to the outcome, and why u choose these measurement as described in fig 1

Validity of the findings

No comments

Additional comments

the topic of the study is new, but the question of the study seems to be with minor clinical relevance

---

## Round 0.2 · Minor Revisions

Please make the final minor revisions before final acceptance.

Reviewer 2 ·

Basic reporting

The language should be improved:
1. Lines 67-68: Add "the" before NPC
2. Line 117: The inclusion criteria describing the missing teeth are not clear. Add "full" before "anterior sextant" and consider to rephrase this section.
3. Lines 119-120: Change "deep overjet" to "increased overjet"
Line 291: Delete the word "the". Change "was" to "were"
Line 338: change "about" to "regarding the"

Experimental design

In the exclusion criteria it was mentioned that extracted teeth with visible residual bone grafts in the CBCT were not included. Therefore, It should be noted in the inclusion criteria that the edentulous sites consisted of extraction with and without socket preservation. Socket preservation may have an impact on bone dimensions after toot extraction and may also influence the measurements performed in this study. A distinction between sites with or without socket preservation should be included in the study, and the influence of socket preservation should be discussed. If this data is unavailable, it should be noted in the discussion as a limitation of the study.

Validity of the findings

No comments

Reviewer 3 ·

Basic reporting

not comments

Experimental design

no comments

Validity of the findings

no comments

Additional comments

I think after the revision, the article meets the criteria, and can be accepted

---

## Round 0.3 · accepted · Accept

Congratulations! Thank you for your revision.